# Analysis of Porcine Pro- and Anti-Inflammatory Cytokine Induction by *S. suis* In Vivo and In Vitro

**DOI:** 10.3390/pathogens9010040

**Published:** 2020-01-03

**Authors:** Florian S. Hohnstein, Marita Meurer, Nicole de Buhr, Maren von Köckritz-Blickwede, Christoph G. Baums, Gottfried Alber, Nicole Schütze

**Affiliations:** 1Institute of Immunology, Center of Infectious Diseases, Faculty of Veterinary Medicine, Leipzig University, Deutscher Platz 5, 04103 Leipzig, Germany; florian.hohnstein@vetmed.uni-leipzig.de (F.S.H.); nicole.schuetze@vetmed.uni-leipzig.de (N.S.); 2Department of Physiological Chemistry, University of Veterinary Medicine Hanover, Foundation, Bünteweg 17, 30559 Hanover, Germany; marita.meurer@tiho-hannover.de (M.M.); nicole.de.buhr@tiho-hannover.de (N.d.B.); maren.von.koeckritz-blickwede@tiho-hannover.de (M.v.K.-B.); 3Research Center for Emerging Infections and Zoonoses (RIZ), University of Veterinary Medicine Hanover, Foundation, Bünteweg 17, 30559 Hanover, Germany; 4Institute of Bacteriology and Mycology, Center of Infectious Diseases, Faculty of Veterinary Medicine, Leipzig University, An den Tierkliniken 29, 04103 Leipzig, Germany; christoph.baums@vetmed.uni-leipzig.de

**Keywords:** *S. suis*, pig, whole-blood model, PBMC, monocytes, interleukin-6, interleukin-10, tumor necrosis factor-alpha, bacterial killing

## Abstract

Weaning piglets are susceptible to the invasive *Streptococcus (S.) suis* infection, which can result in septicemia. The aim of this study was to investigate the cytokine profile induced upon *S. suis* infection of blood, to determine the cellular sources of those cytokines, and to study the potential effects of the induced cytokines on bacterial killing. We measured TNF-α, IL-6, IFN-γ, IL-17A and IL-10 after an experimental intravenous infection with *S. suis* serotype 2 in vivo, and analyzed whole blood, peripheral blood mononuclear cells (PBMC) and separated leukocytes to identify the cytokine-producing cell type(s). In addition, we used a reconstituted whole blood assay to investigate the effect of TNF-α on bacterial killing in the presence of different *S. suis*-specific IgG levels. An increase in IL-6 and IL-10, but not in IFN-γ or IL-17A, was observed in two of three piglets with pronounced bacteremia 16 to 20 h after infection, but not in piglets with controlled bacteremia. Our results confirmed previous findings that *S. suis* induces TNF-α and IL-6 and could demonstrate that TNF-α is produced by monocytes in vitro. We further found that IL-10 induction resulted in reduced secretion of TNF-α and IL-6. Rapid induction of TNF-α was, however, not crucial for in vitro bacterial killing, not even in the absence of specific IgG.

## 1. Introduction

*Streptococcus (S.) suis* is a challenging problem in pig breeding with zoonotic potential [1,2,3]. Weaning piglets are commonly asymptomatically colonized with *S. suis* at mucosal surfaces [4,5]. Bacteria can be found shortly after birth in several locations, including the saliva [6], palatine and nasopharyngeal tonsils and mandibular lymph nodes [7,8]. However, when *S. suis* becomes invasive, it can induce severe diseases associated with meningitis, endocarditis, arthritis or septicemia [9]. An acute disease outbreak can lead to a loss of 4–12% of weaning piglets [7,10]. To date, *S. suis* has been classified into various serotypes [11,12,13,14,15,16,17,18]. Prevalent serotypes causing clinical *S. suis* cases in swine in Europe are strains with the capsular polysaccharide synthase (*cps) 9*, *cps2*, and *cps7* [1]. The bacterial capsule of *S. suis* has been shown to play an important role during infection, most notably in protection against phagocytosis and killing by neutrophils and dendritic cells [19,20,21,22,23,24,25,26]. The induction of pro-inflammatory cytokines like TNF-α, interleukin (IL)-1β, IL-6 or IL-8 by the bacterial cell wall components has been shown for murine macrophages [27] and human THP-1 monocytes [28]. Another study in human monocyte-derived dendritic cells showed that *S. suis cps2* was able to modulate cytokine production towards a more anti-inflammatory profile compared to other serotypes [24]. In addition, the induction of pro-inflammatory cytokines by *S. suis* in porcine blood has been investigated in vitro [29], but in vivo data on the relationship between induced cytokines and bacteremia in pigs are missing.

The aim of this study was to analyze the cytokine response to *S. suis* infection in the blood compartment to understand how bacteremia in piglets is linked with the release of pro- and anti-inflammatory cytokines. Furthermore, we wished to compare the in vivo situation of bacteremia with in vitro models, and to investigate the potential effects of cytokines on bacterial killing. Specifically, we analyzed the serum levels of TNF-α, IL-6, IFN-γ, IL-17A, and IL-10 in pigs intravenously infected with *S. suis.* Based on in vivo data, the cytokine production in whole blood and PBMC was analyzed, and cellular sources of the induced cytokines were evaluated. Furthermore, by concurrent analysis of *S. suis* survival, as well as TNF-α neutralization or addition of recombinant TNF-α, we investigated the potential effects of cytokine release on bacterial killing in blood.

## 2. Results

### 2.1. Detection of IL-6 and IL-10 in Sera of Piglets with Pronounced Bacteremia after Intravenous Infection with S. suis

The induction of inflammatory cytokine responses upon streptococcal infection has been demonstrated in previous studies in human whole blood and also by bacterial stimulation in murine in vitro models [27,30,31]. Previously, *S. suis* has been shown to induce pro-inflammatory cytokines in a study with porcine whole blood [29]. In this study, we investigated whether *S. suis* induces cytokines during bacteremia in vivo after an intravenous infection that mimics bacteremia caused by an invasive infection. We determined cytokine levels in serum samples taken from nine piglets (weaned at four to five weeks of age; six to ten weeks old at time of analysis) at 0, 13, 16, and 19 h (time points given by the animal permit), six of which were intravenously infected with *S. suis* strain 10 and three of which served as uninfected controls. To prevent stress-induced immunomodulation during blood sampling, all experimental animals were under anesthesia from 13 to 19 h post infection when blood was drawn. Three of six infected individuals developed pronounced bacteremia 13 h post infection (hpi; Figure 1A). Two of three pigs with pronounced bacteremia showed an increase in serum IL-6 (Figure 1B): individual H7 showed an IL-6 peak of 1.9 ng/mL at 16 hpi, and individual H2 a peak of 0.9 ng/mL at 19 hpi. For IL-10, an increase of serum levels with pronounced bacteremia was also detectable in two of three piglets: piglet H2 showed an increase up to 1.6 ng/mL at 19 hpi. Piglet H3, which did not show elevated IL-6 levels, showed an IL-10 peak of 0.7 ng/mL at 16 hpi. The piglet H7 did not show increased IL-10 secretion. In contrast, no increase in IL-6 or IL-10 levels was detectable in the sera of three pigs, which were able to control bacteremia, as well as in the three uninfected control pigs. In addition, TNF-α, IL-17A and IFN-γ were measured in the serum of all experimental animals, but were not detectable during this time frame (Appendix A).

In summary, we found that during infection, cytokines are induced in vivo in piglets with pronounced *S. suis* serotype 2 bacteremia. However, it remained open whether TNF-α was induced at earlier time points after infection.

### 2.2. In Vitro Induction of Pro- and Anti-Inflammatory Cytokines by S. suis is Limited by IL-10

As we observed an induction of IL-6 and IL-10 in piglets with pronounced bacteremia, we tested whether cytokine induction could be reproduced in a whole-blood assay upon addition of viable *S. suis*, as has been demonstrated previously [29]. Since IL-6 and TNF-α are commonly induced immediately after infection [27,29], we measured cytokines in plasma at earlier time points (2 and 6 h) after stimulation with *S. suis* than determined in the in vivo-infection experiment. To assess the proliferation or killing of *S. suis* in the whole-blood assay, bacterial survival was determined after 2 and 6 h. Whole blood of eight to nine week old piglets was stimulated with the same bacterial strain from the infection experiment (strain 10, *cps2*). As a further control, we used an acapsular mutant of the encapsulated wt strain 10, described as 10cpsΔEF [19], that is known to show an enhanced cytokine release by *S. suis* [29,32], since the capsule covers the immuno-stimulatory cell wall components. We found that strain 10 induced TNF-α and IL-6 after 2 h and at significantly increased levels after 6 h, while IL-10 was only induced 6 h after stimulation (Figure 2A). The magnitude and time course of the cytokine induction for the acapsular strain was very similar. In contrast to similar cytokine responses, the acapsular strain was killed after 2 h, whereas the wild type (wt) strain 10 proliferated within 2 and 6 h after infection (Figure 2B).

In summary, we confirmed the in vivo data in a whole-blood assay (Figure 1) by showing that *S. suis* strain 10 induces pro-inflammatory (i.e., IL-6) and anti-inflammatory (i.e., IL-10) cytokines in blood. In addition, we detected an early TNF-α induction (2 h). We could also confirm previous findings [29] that acapasular *S. suis* induces similar cytokine levels, but shows a reduced survival in a porcine whole blood.

Bacterial proliferation and/or neutrophil-mediated killing of *S. suis* in the used whole-blood assay can change the ratio of bacteria to blood cells and complicate the interpretation of data. Thus, to exclude bacterial proliferation and to maintain a defined ratio of bacteria to blood cells in the following experiments, we used isolated PBMC in a cell culture system with antibiotics to analyze *S. suis*-induced cytokine production. To prevent phagocytosis and killing as much as possible we used PBMC, which no longer contain granulocytes, since granulocytes are important for bacterial killing [33]. PBMC, isolated from healthy weaning piglets, were stimulated with the wt *S. suis* strain 10 and the acapsular mutant. In PBMC, we observed that the induction of TNF-α, IL-6 and IL-10 by the wt strain was dependent on the bacterial load, but resulted in only marginal cytokine levels (Figure 3A). In contrast to wt *S. suis*, cytokine induction after stimulation with the acapsular strain was significantly higher, but IL-6 and IL-10 also showed a dose-dependent increase, whereas TNF-α was already strongly induced at a PBMC-to-bacteria ratio of 10:1.

In summary, we observed cytokine induction by stimulation with the wt strain, dependent on the high bacterial load and dose-dependent induction of IL-6 and IL-10 by the acapsular strain. The observed dependency on the bacterial load is in line with our previous findings (Figure 2), since a high bacterial load of the wt strain 6 h upon infection (Figure 2B) was associated with higher cytokine induction (Figure 2A). Moreover, the association found in vivo for cytokine production and pronounced bacteremia (Figure 1) underlines the bacterial dose-dependency observed in vitro using either whole blood or PBMC.

To test if the results obtained using the *cps2* strain 10 are representative of different *S. suis* serotypes, we included two further *S. suis* strains with clinical relevance, but different serotypes: the *cps7* strain 13-00283-02 [34] and the *cps9* strain 16085/3b [35]. TNF-α was induced at similar levels by all three strains and increased dose-dependently up to 6 ng/mL (Figure 3B). IL-6 induction was highly donor-dependent and varied between 0 and 4 ng/mL in all three strains. The IL-10 induction was also dependent on the bacterial load between 0.1 and 1.8 ng/mL by all three strains. This shows that *S. suis* strains of different serotypes induce TNF-α, IL-6 and IL-10 similarly.

IL-10 is a potent anti-inflammatory mediator. To investigate a possible regulatory effect of *S. suis*-induced IL-10 on the production of the pro-inflammatory cytokines IL-6 or TNF-α, both cytokines were measured following *S. suis* stimulation of PBMC in the presence of a neutralizing anti-porcine IL-10 antibody. Compared to the control, IL-10 neutralization increased the levels of TNF-α and IL-6 (Figure 3C). Although the differences were not significant, we found a trend comparing the PBMC of individual piglets in the absence and presence of neutralizing anti-IL-10 antibodies.

These findings suggest that *S. suis*-induced IL-10 might reduce the production of the pro-inflammatory cytokines IL-6 and TNF-α.

### 2.3. Encapsulated S. suis Induces TNF-α in Monocytes, but Not in Granulocytes

Cells of the innate immune system, such as monocytes, macrophages and neutrophils, have been shown to be producers of TNF-α, IL-6 and IL-10 upon stimulation with bacterial antigens in mice and humans [28,30,36,37]. To determine the contribution of myeloid cells of the blood to the production of the detected cytokines, we stimulated porcine PBMC, CD14-positive monocytes isolated from the PBMC and granulocytes from the blood of pigs from the same herd with wt *S. suis* strain 10 or with the acapsular mutant, and measured cytokines in the supernatants. The blood leukocytes of pigs contain up to 50% granulocytes and 5–10% monocytes [38]. In vitro cell numbers of PBMC, CD14-positive monocytes and granulocytes were chosen accordingly: PBMC—1 × 10^6^ cells/well (200 µL), monocytes—5 × 10^4^ cells/well, granulocytes—1 × 10^6^ cells/well. CD14-positive monocytes can also be detected by flow cytometry using CD172a staining for separated PBMC (Appendix A). For adequate comparison of PBMC, isolated monocytes and granulocytes, the same *S. suis* concentration of 1 × 10^6^ colony-forming units (cfu)/well was used to stimulate all cell fractions (equivalent to a 1:1 ratio of unseparated PBMC to bacteria). 

The PBMC fraction, containing lymphoid cells and monocytes, produced low amounts of TNF-α (median 0.3 ng/mL), IL-6 (median 0.1 ng/mL) and IL-10 (median 0.2 ng/mL) in response to wt *S. suis,* and high amounts (median of TNF-α: 14.5 ng/mL, IL-6: 1.0 ng/mL, IL-10: 1.7 ng/mL) in response to the acapsular mutant (Figure 4A). The monocyte fraction produced clearly detectable levels of TNF-α (median 0.7 ng/mL) after stimulation with the wt strain and high levels of TNF-α (median 4.0 ng/mL) with the acapsular strain. IL-6 and IL-10 were detectable in monocytes stimulated with the acapsular strain (median IL-6: 0.4 ng/mL, IL-10: 0.6 ng/mL), but not after stimulation with the wt strain. Neither IL-6 nor IL-10 were detectable in granulocytes when stimulated with the wt strain (Figure 4B). Stimulation with the acapsular strain induced TNF-α (max. 3.3 ng/mL, median 0.42 ng/mL) in granulocytes, but at lower levels than in stimulated monocytes, whereas IL-6 and IL-10 induction in granulocytes was detectable only at a very low level in some samples (Figure 4B).

In summary, we could demonstrate that monocytes are the main myeloid producers of TNF-α induced by encapsulated wt *S. suis*.

### 2.4. TNF-α Induction is Not Crucial for Bacterial Killing of S. suis In Vitro, Not Even in the Absence of S. suis-Specific Immunoglobulin (Ig) G

We observed that *S. suis* induces pro- and anti-inflammatory cytokines in vivo (Figure 1), in whole blood (Figure 2) as well as in PBMC and in isolated monocytes (Figure 3 and Figure 4). The most prominent cytokine detected in our in vitro assays is TNF-α, since it is produced in high amounts in the whole-blood assay even by the encapsulated wt strain 10 (Figure 2). Whether *S. suis*-induced TNF-α is able to affect bacterial killing remains elusive.

We observed that TNF-α secretion was induced rapidly after bacterial contact (2 h, Figure 2A). To determine the time point of bacterial killing and TNF-α release more precisely, we performed a kinetic analysis in whole blood in 30 min intervals after in vitro infection with wt *S. suis* strain 10 (2 × 10^6^ cfu/mL). TNF-α levels in plasma and bacterial survival factors were determined at these time points. We found that TNF-α was induced only after 90 min of stimulation, but bacteria were already killed after 30 min (Figure 5A).

Detection of *S. suis* (strain 10)-specific IgG from plasma samples of the whole blood revealed a high antibody background (>60 RU/mL, defined as “IgG-high”, Figure 5C) in these blood samples. This suggests that TNF-α does not contribute to bacterial killing in this experimental setup, but that *S. suis*-specific antibodies are critical for the rapid killing. Therefore, we used whole blood samples with low *S. suis*-specific IgG (<60 RU/mL, defined as ”IgG-low”, Figure 5C) and, in a further approach, we reconstituted blood cells with colostrum-deprived serum (CDS, named “no IgG”) to completely exclude the interfering effects of antibody-mediated mechanisms on bacterial killing. To investigate the effects of TNF-α in more detail, we added recombinant TNF-α (rTNF-α, 10 ng/mL) or a neutralizing anti-TNF-α antibody to the *S. suis*-stimulated blood assay. In whole blood of the IgG-low group, the addition of rTNF-α did not increase bacterial killing (Figure 5B, left panel). To investigate the effect of endogenous TNF-α on killing, a neutralizing anti-TNF-α antibody was additionally used. The neutralization of endogenous TNF-α also did not change bacterial survival. In the approach with reconstituted porcine blood cells with CDS, *S. suis* induced TNF-α only marginally (Figure 5B, right panel). As expected, *S. suis* showed better survival in the CDS background compared to the IgG-low background. However, bacterial killing was not affected by the addition of rTNF-α.

We conclude that under the chosen experimental conditions, TNF-α does not contribute to the killing of *S. suis* in blood, even in the absence of specific antibodies.

## 3. Discussion

*S. suis* is an important pathogen in the piglet farming industry. The pathogenesis of *S. suis* is linked to invasive infection resulting in septicemia, meningitis, arthritis, or endocarditis [8]. Bacteremia as a potential mechanism for dissemination of this pathogen goes along with the manifold clinical symptoms. Whether bacteremia provokes the release of cytokines in the natural host of *S. suis* was the focus of this study by an experimentally induced invasive infection.

We could detect an induction of IL-6 and the anti-inflammatory mediator IL-10 in serum of intravenously *S. suis*-infected piglets (after 13–19 hpi) that showed a pronounced bacteremia. By in vitro stimulation with *S. suis,* we confirmed our in vivo results as well as previous results by Segura et al. [29], that *S. suis* induces TNF-α and IL-6 following in vitro stimulation of porcine blood. Similar to our in vivo data, increased cytokine production accompanies the growth of *S. suis* in the whole-blood assay and in PBMC stimulated with higher doses of bacteria. Thus, our data demonstrate that increased bacterial load with *S. suis* in the blood leads to increased cytokine production. This shows the relevance of the whole-blood assay as an in vitro system and confirms earlier findings [29] using a whole-blood assay that can mimic some aspects of bacteremia. Furthermore, the study demonstrates the survival and growth of *S. suis* in blood despite the presence of pro-inflammatory cytokines IL-6 and TNF-α. Therefore, these pro-inflammatory cytokines do not appear to be crucial for killing mechanisms. Specifically, TNF-α did not play a role in bacterial killing in whole blood in vitro when recombinant TNF-α or a neutralizing antibody were added, not even in conditions with low *S. suis*-specific antibodies. In addition, we show that the induction of IL-10 limits the production of the pro-inflammatory cytokines TNF-α and IL-6. 

The induced cytokines detected in our study (Figure 1 and Figure 2) match the pro-inflammatory cytokine profile described in the in vitro study in porcine blood by Segura et al. [29], as well as the comparable cytokine levels, by stimulation with different *S. suis* strains (Figure 3B) [29]. The differences found in the onset and magnitude of cytokine production in vitro may be related to genetic differences between the pig breeds and differences in procedures of in vitro experiments. The sequential onset of cytokine production, starting with TNF-α and followed by IL-6 and IL-10, has also been found in other studies [29,32,39,40]. A systemic infection (intraperitoneally) with a suilysin-positive virulent serotype 2 strain in CD1 mice led to a detectable TNF-α increase in serum between 3 and 6 h, but to a decline at 6 h, similar to TNF-α levels before infection [31]. Due to experimental restrictions by the animal permit for the in vivo infection experiment (Figure 1), the analysis of serum cytokine levels between 1 and 13 h post infection was not possible. Therefore, an early peak of TNF-α (1 to 13 h) is conceivable.

For the human monocyte cell line THP-1, the release of pro-inflammatory cytokines, like IL-6 and TNF-α, in response to *S. suis,* was demonstrated [28]. However, a human monocyte cell line is probably not comparable with the situation in porcine circulation. We separated monocytes from peripheral blood to characterize the source of cytokines using anti-human CD14 beads. Monocytes could be identified as the main producers of TNF-α by comparing PBMC to isolated monocytes after the addition of *S. suis* (Figure 4). Monocytes possess numerous pattern recognition receptors and could be more reactive to *S. suis* due to a previously described extracellular binding of *S. suis* with monocytes in the blood stream, independent of phagocytosis [21,41]. The bacterial attachment to phagocytic cells without phagocytosis was also demonstrated for the murine macrophage cell line J774 [42]. Furthermore, we observed a strongly reduced TNF-α production by the reconstitution of blood with CDS as compared with sera containing high and low levels of *S. suis*-specific IgG (Figure 5). Granulocytes, on the other hand, only secreted TNF-α when stimulated with the acapsular mutant strain allowing for facilitated uptake, which suggests that phagocytosis may be associated with cytokine induction in granulocytes, since the phagocytosis of *S. suis* is impaired by the polysaccharide capsule [21].

In the whole-blood assay, the acapsular strain induced cytokines at a high magnitude although it was killed rapidly, while the wt strain induced similar cytokine levels, but was not killed (Figure 2). The wt strain can evade killing through its bacterial capsule, which has been shown to effectively inhibit phagocytosis [20,21,22,25]. The higher proliferation of the wt strain results in an increased and prolonged exposure of immune cells to bacterial antigens. However, the capsule does not only mask surface-associated bacterial antigens, but exposed cell wall components are also more potent cytokine inducers than the capsule [29,32]. Previous studies in whole blood and human cell culture have demonstrated that cell wall components of *S. suis* can induce TNF-α and IL-6, depending on the CD14 receptor [28] and on the toll-like receptor (TLR) 2 [32]. Therefore, the acapsular strain induces cytokines at a higher magnitude [29,32].

The current understanding is that killing of *S. suis* is dependent on phagocytosis [33] and mediated by opsonizing antibodies. Segura et al. showed that opsonization of the *S. suis* serotype 2 with specific immune serum reduced induction of the pro-inflammatory cytokines TNF-α, IL-1β and IL-6, associated with decreased bacterial survival [29]. The observed in vivo induction of IL-6 and IL-10 in bacteremic pigs (Figure 1) raises the question of whether the induction of pro-inflammatory cytokines also has protective properties in bacteremia. However, it has been shown that decreasing IL-6 levels are associated with a protective effect in pneumococcal septicemia in pigs [43]. Furthermore, a co-infection study on porcine bone marrow-derived dendritic cells (BMDC) experimentally infected with porcine reproductive and respiratory syndrome virus (PRRSV) and a secondary infection with *S. suis* showed that phagocytosis of bacteria was impaired when IL-6 and TNF-α were upregulated [44], pointing to a detrimental effect of IL-6 or TNF-α upregulation. Therefore, we suggest that the induction of IL-10 by *S. suis* and the observed limiting effect of IL-10 on TNF-α and IL-6 secretion (Figure 3C) can be interpreted as a physiological negative feedback to counteract an uncontrolled pro-inflammatory immune response, and to ultimately allow for reversion to a state of immunological equilibrium. However, this feedback can have detrimental effects, too. The infection of mice with influenza virus has been shown to trigger increased IL-10 levels in reconvalescent animals, but ultimately resulted in an increased susceptibility to *S. pneumoniae* infection and a lethality of 100% [45]. Since it has been shown that porcine PBMC also highly upregulates IL-10 after PRRSV infection [46], a similar scenario could be possible in the field in pigs infected with PRRSV and a secondary invasive *S. suis* infection. The detrimental effects of high IL-10 induction have been further demonstrated for *Klebsiella pneumoniae* [47] and meningococcal infection [48], and could result in reduced phagocytic and bactericidal activity in neutrophils [49]. 

With regards to TNF-α downregulation by IL-10, our data demonstrate that a potential negative impact of IL-10 on *S. suis* killing is unlikely, since we could not find TNF-α involved in bacterial killing in vitro. Since TNF-α is also known as an inducer of oxidative burst in neutrophils, this cytokine could be able to support antibody-mediated killing mechanisms against the catalase-negative *Streptococci* [50]. To this end we, however, found inconsistent (donor-dependent) induction of an oxidative burst by the addition of 10 ng/mL TNF-α (data not shown). On the other hand, TNF-α may contribute to an inflammatory response detrimental to the host. However, in a pig model of *S. pneumoniae* sepsis, it was directly demonstrated that neutralization of TNF-α has no protective anti-inflammatory effect [51]. In a recent study, the inhibition of an inflammasome-triggered cytokine storm in an *S. suis* mouse model did not influence the bacterial load, but reduced mortality and clinical score [52]. For a more detailed understanding of the role of cytokines during an invasive infection in pigs, histopathological scoring could be of advantage to investigate the local detrimental effects induced by pro-inflammatory cytokines. 

In summary, a high bacterial antigen load seems to be associated with the induction of IL-6 and TNF-α, and could cause an inflammatory cascade during bacteremia, potentially contributing to a detrimental outcome of invasive *S. suis* infection. Specifically, TNF-α does not contribute to the bacterial killing in whole blood in vitro. The concurrent induction of IL-10 in the blood could be a possible regulatory mechanism to control the release of pro-inflammatory cytokines.

## 4. Methods

### 4.1. Bacterial Strains

The *cps2* strain 10 [53] and the isogenic capsule mutant strain 10cpsΔEF [19] were kindly provided by Hilde Smith (Wageningen, GE, The Netherlands). The *cps7* strain 13-00283-02 and the *cps9* strain 16085/3b have been characterized previously in whole-blood killing assays and confirmed to be virulent in experimental infections with piglets [34,35].

Strains were grown at 37 °C in Todd-Hewitt broth (THB, Becton Dickinson GmbH, Heidelberg, BW, Germany) until the late exponential growth phase and stored at −80 °C in 20% sterile glycerol until usage for whole-blood assays, or washed two times with PBS in case of usage for PBMC stimulation, before storage at −80 °C in 20% glycerol.

For the experimental infection of pigs, *S. suis cps2* strain 10 was grown in Tryptic Soy Broth without Dextrose (TSB; Becton Dickinson GmbH, Heidelberg, BW, Germany, product #286220) at 37 °C with 5% CO_2_. Infection inoculum were prepared at the late exponential growth phase.

### 4.2. Animal Experiments and Blood Sampling

All piglets used in this study were from conventional herds with carriers of numerous *S. suis* genotypes. Whole blood samples for in vitro experiments were drawn from *Vena cava cranialis* from healthy piglets between 6 and 10 weeks of age (German Landrace, weaned at four weeks of age); in 9 mL Li-heparin tubes (Sarstedt AG&Co, Nümbrecht, NW, Germany). Porcine blood was taken in accordance with the permit no. N19/14, approved by the responsible authorities of the state of Saxony, Germany (Landesdirektion Sachsen, Chemnitz, SA, Germany).

In vivo data were generated from samples of an independent infection experiment carried out at the University of Veterinary Medicine Hannover, Germany (permit No. 33.8-42502-04-18/2879, Committee on Animal Experiments of the Lower Saxonian State Office for Consumer Protection and Food Safety) with German Landrace piglets from a different herd. The handling and treatment of animals was in strict accordance with the principles of the European Convention for the Protection of Vertebrate Animals Used for Experimental and Other Scientific Purposes, as well as the German Animal Protection Law. Eight week-old German Landrace piglets (weaned at four to five weeks of age) were intravenously infected under ketamine and azaperone anesthesia with 3 × 10^8^ cfu of *S. suis cps2* strain 10 [53]. The goal of the infection was to induce meningitis, but no arthritis or shock symptoms. The infection dose was based on experience with previous intranasal and intravenous infection studies in piglets of the same age group with *S. suis* serotype 2 and serotype 9, respectively. PBS was intravenously injected in the control animals under same conditions. Twelve hours post infection, the piglets were anaesthetized through application of ketamine and azaperone, and anesthesia was maintained for six hours (i.e., from 13 to 19 h after intravenous infection) via inhalation of isoflurane while blood was drawn. Blood samples were taken from *Vena jugularis* or *Vena cava cranialis* pre-infection and from *Arteria femoralis*, respectively, at defined time points (0, 13, 16 and 19 h post infection) and filled into serum and plasma (lithium–heparin) monovettes (SARSTEDT AG & Co. KG, Nümbrecht, NW, Germany). From Li-heparin blood, serial dilutions were plated on Columbia blood agar and the bacterial load (cfu/mL) was determined after incubation at 37 °C for 20 h. In addition, serum samples were taken, centrifuged 30 min after blood withdrawal, and immediately frozen after centrifugation in small volumes in liquid nitrogen and stored at −80 °C until usage.

Colostrum-deprived serum was obtained from piglets that were born dead without showing signs of decay or from very weak, newborn piglets that were euthanized immediately after birth due to the inability to rise and to take up colostrum.

### 4.3. Bactericidal Assay

Bactericidal assays were conducted as previously described [54] with some modifications. Briefly, 500 µL heparinized blood was mixed with *S. suis* strains at a concentration of 2 × 10^6^ cfu/mL (time point t_0_) and incubated at 37 °C for 2 or 6 h on a rotator at eight rounds per minute (rpm).

To study the time course of TNF-α induction and bacterial survival, whole blood from each piglet was mixed at room temperature with *S. suis* for a final concentration of 2 × 10^6^ cfu/mL. Infected blood was immediately distributed to separate reaction tubes for each time point (0, 30, 60, 90, and 120 min; 500 µL each) and incubation at 37 °C on a rotator (8 rpm) was started simultaneously for all tubes.

To calculate survival factors, blood was diluted and incubated on Columbia agar in duplicate (running droplet). Cfu were counted after overnight incubation at 37 °C. The survival factor at a given time point t_x_ was calculated as quotient of cfu at t_x_ divided by cfu at t_0_.

For functional studies, recombinant porcine TNF-α (10 ng/mL) or neutralizing anti-TNF-α antibody (4 µg/mL; both R&D Systems Inc., Minneapolis, MN, USA) was added. Bioactivity of recombinant TNF-α has been demonstrated previously [55].

For a serum-reconstituted bactericidal assay, whole blood was washed twice with PBS and afterwards supplied with half the initial blood volume of defined sera (e.g., from colostrum-derived pigs).

### 4.4. Isolation of PBMC, Monocytes, and Granulocytes

PBMC were isolated from whole blood by density gradient centrifugation with Biocoll (1.077 g/mL; Biochrom/Merck KGaA, Darmstadt, BW, Germany).

For magnetic cell separation of monocytes, PBMC were labeled with anti-human CD14 microbeads and separated via MACS^®^ MS columns (Miltenyi Biotec GmbH, Bergisch Gladbach, NW, Germany) in the magnetic field of a miniMACS™ separator. Monocyte purity was determined by flow cytometry analysis, as detailed below.

Granulocytes were collected from the pellet of a density gradient separation of whole blood with Biocoll (described above), and again subjected to a second Biocoll density gradient separation. Erythrocytes were removed by incubation in erythrocyte lysis buffer (0.155 M ammonium chloride, 10 mM potassium bicarbonate, 0.1 mM disodium EDTA) for 5 min on ice. To determine the purity of granulocytes, the separated fraction was transferred to glass slides (2 × 10^5^ cells/slide) with a Cytospin™ 4 centrifuge (Thermo Fisher Scientific, Waltham, MA, USA, 3 min at 1000 rpm) and stained with a Diff Quik^®^ staining kit (Medion Diagnostics AG, Düdingen, FR, Switzerland). The frequencies of lymphocytes, monocytes and granulocytes (neutrophils, eosinophils and basophils) out of 100 total cells were determined microscopically. The purity of granulocytes determined with this method was 96% (median, Appendix A). The viability of granulocytes was confirmed by cell count in 0.4% trypan blue solution.

### 4.5. Cell Culture and Stimulation

Isolated PBMC were seeded into sterile 96-well flat-bottom plates at 10^6^ cells/well in 200 µL Iscove’s Modified Dulbecco’s Medium (IMDM; PAN-Biotech GmbH, Aidenbach, BY, Germany) supplemented with 10% fetal bovine serum (FBS, PAN-Biotech GmbH, Aidenbach, BY, Germany), 50 µg/mL gentamicin (Merck KGaA, Darmstadt, BW, Germany), 100 U/mL penicillin, and 100 µg/mL streptomycin (Pen/Strep; Merck). Viable *S. suis* strains from glycerol stocks were added at PBMC-to-bacteria ratios of 10:1 or 1:1 (corresponding to 10^5^ and 10^6^ cfu/well), as indicated in the respective figure descriptions and legends.

For the separation experiments, PBMC were again cultivated at 10^6^ cells/well (200 µL). CD14-positive monocytes were cultivated at 5 × 10^4^ cells/well (5% of PBMC); granulocytes were cultivated at 1 × 10^6^ cells/well. For stimulation, the same final concentration of 10^6^ cfu/well was used for all three fractions (equivalent to a 1:1 ratio of PBMC to bacteria).

For IL-10 neutralization, anti-porcine IL-10 (clone 148801; R&D Systems Inc.) or isotype control mIgG2b (clone 20116; R&D Systems Inc.) were added at 0.1 µg/mL, respectively. Bioactivity of clone 148801 has been demonstrated previously [56]. After 42 h of stimulation with *S. suis*, supernatants were taken and stored at −20 °C.

### 4.6. Flow Cytometry

Separated monocytes were stained with viability dye eFluor™ 506 (Thermo Fisher Scientific, Waltham, MA, USA), anti-CD3 (clone BB23-8E6-8C8; Becton Dickinson GmbH, Heidelberg, BW, Germany), and anti-CD172a (“SWC3”, clone 74-22-15A; Becton Dickinson GmbH) to validate purity. Flow cytometric measurements were done on an LSR Fortessa (Becton Dickinson GmbH). Flow cytometric data were analyzed with FlowJo software 10.1r5 (TreeStar, Ashland, OR, USA).

### 4.7. Cytokine Quantification

ELISA kits for porcine IL-6, IL-10, and TNF-α were purchased from R&D Systems Inc. Ninety-six-well plates (Nunc MaxiSorp™ round-bottom; Thermo Fisher Scientific) were coated with antibodies according to the manufacturer’s instructions. The biotinylated detection antibodies were coupled with a streptavidin–horseradish peroxidase conjugate. The peroxidase was developed with a 3,3’,5,5’-Tetramethylbenzidin (TMB) solution (SeraCare, Milford, MA, USA, formerly KPL) and the reaction was stopped after 20 min with H_3_PO_4_. Optical density (OD) values were measured with a microplate reader SpectraMax 340PC384 (Molecular Devices, LLC San Jose, CA, USA) at 450 and at 630 nm as a background reference, and analysed with SoftMax^®^ Pro v5.0 software (Molecular Devices, LLC).

### 4.8. S. suis-Specific IgG Quantification

Round-bottom 96-well plates (Nunc MaxiSorp round-bottom; Thermo Fisher Scientific) were coated with *S. suis* strain 10 and blocked with 5% skim milk solution. After incubation with control and sample sera, IgG was detected with a goat anti-pig IgG antibody coupled to peroxidase. The peroxidase was developed with a citrate buffer containing 0.003% H_2_O_2_ and 500 µg/mL of the 2,2’-azino-bis(3-ethylbenzothiazoline-6-sulphonic acid) (ABTS) substrate. OD values were measured with a Synergy™ H1 microplate reader (BioTek Instruments, Inc., Winooski, VT, USA) at 450 and at 630 nm as background reference. IgG was quantified in relative units (RU) as a ratio of the samples relative to a control serum, as described previously [57].

### 4.9. Statistics

Data analyses were performed with GraphPad Prism 5 (GraphPad Software, Inc., La Jolla, CA, USA) to calculate normal distribution and statistical significance (α = 0.05) with Student’s *t*-test, Mann-Whitney test, or with Kruskal–Wallis test and Dunn’s post-hoc test.

## 5. Conclusions

*S. suis* induces pro-inflammatory IL-6 and anti-inflammatory IL-10 in bacteremic pigs. IL-10 is able to downregulate pro-inflammatory cytokine induction in vitro. The data suggest that monocytes are the main producers of *S. suis*-induced TNF-α in blood. However, cytokine induction is not associated with the killing of *S. suis*. Specifically, TNF-α does not contribute to bacterial killing in whole blood in vitro, not even in the absence of *S. suis*-specific IgG.

## Figures and Tables

**Figure 1 pathogens-09-00040-f001:**
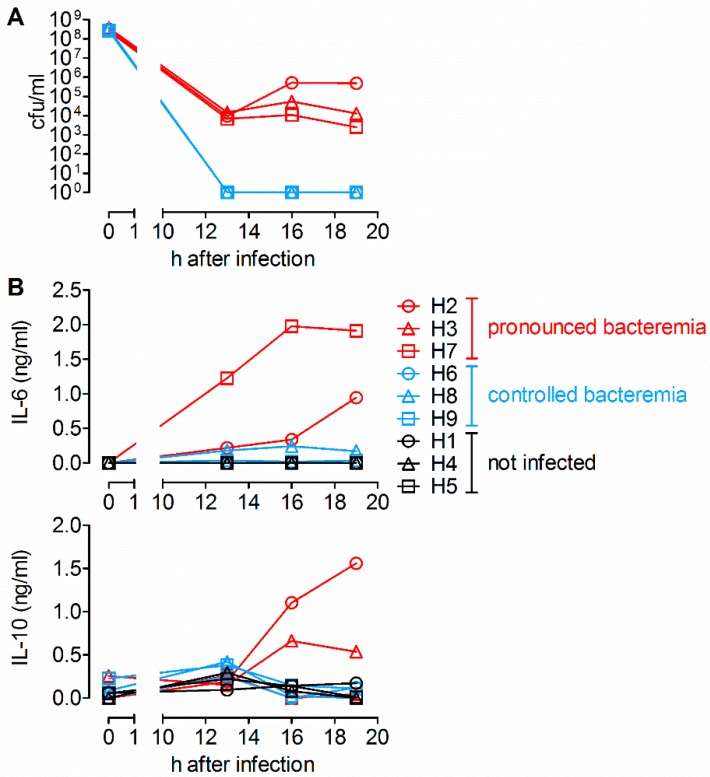
Detection of IL-6 and IL-10 post intravenous *S. suis* infection in sera of piglets with pronounced bacteremia. (**A**) Quantification of bacterial load in blood 0, 13, 16, and 19 h after intravenous infection of six piglets (eight weeks old, weaned at four weeks of age).‡ (**B**) Cytokine levels in individual sera measured by ELISA before and after infection with *S. suis* strain 10 (*cps2*, *n* = 6) and in serum of non-infected control animals (*n* = 3). Note that earlier time points than 13, hpi could not be taken due to the experimental setup. ‡ Some of the bacterial load data shown here were additionally used in another study, though in context with other parameters (manuscript accepted, *Infect Immun*, 3 December 2019).

**Figure 2 pathogens-09-00040-f002:**
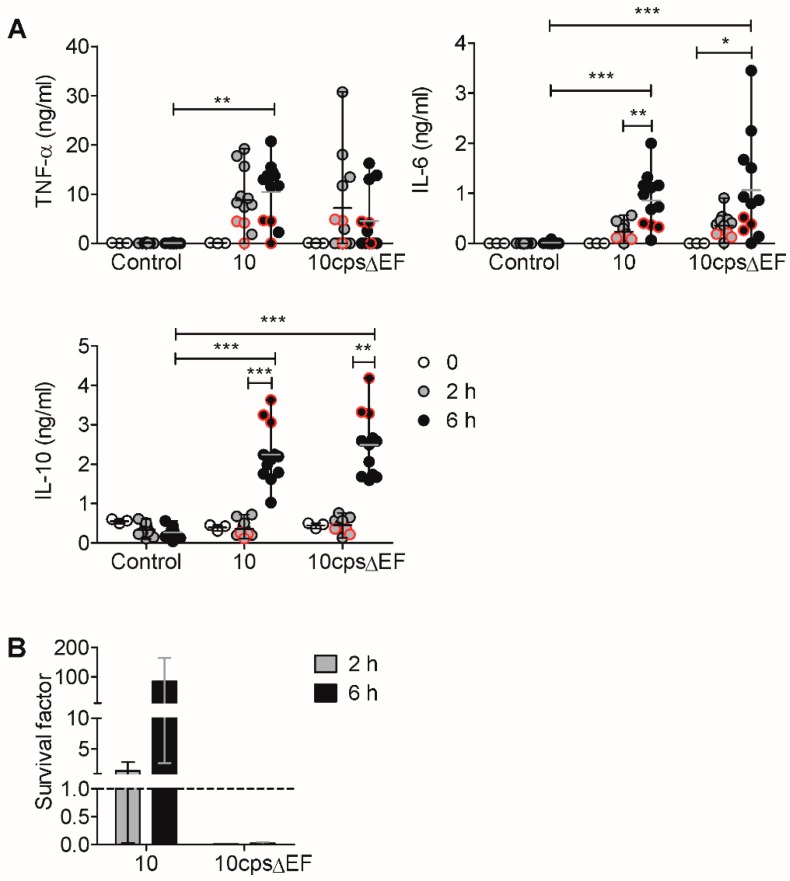
The induction of pro- and anti-inflammatory cytokines by encapsulated and acapsular *S. suis* strain 10 in vitro in whole blood is similar, but bacterial survival is different. (**A**) Quantification of cytokines in plasma before (0 h, *n* = 3) and 2 or 6 h after (*n* = 12) the addition of 2 × 10^6^ colony-forming units (cfu)/mL *S. suis* strain 10 or 10cpsΔEF to whole blood. Median (horizontal line) of all samples are shown; red symbols indicate the individuals that are shown in B. (**B**) Bacterial survival from the same samples 2 and 6 h after in vitro infection. Calculation of survival factors (t_x_) = (cfu at t_x_)/(cfu at t_0_). Bars show median and range of individually calculated survival factors (*n* = 3). Graphs show data of two independent experiments with eight to nine week old piglets. Statistical analysis was conducted by Kruskal–Wallis test and Dunn’s post-hoc test (* *p* < 0.05, ** *p* < 0.01, *** *p* < 0.001).

**Figure 3 pathogens-09-00040-f003:**
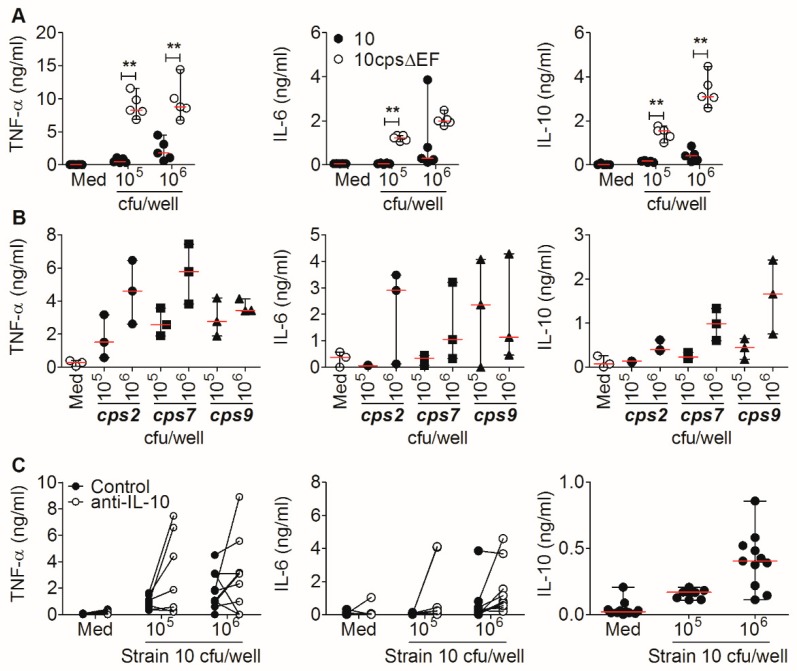
Cytokine release by peripheral blood mononuclear cells (PBMC) is *S. suis* dose-dependent, and IL-6 and TNF-α secretion is limited by IL-10. (**A**) Isolated porcine PBMC were stimulated with strain 10 or the acapsular mutant strain 10cpsΔEF at a PBMC-to-bacteria ratio of 10:1 (10^5^ cfu/well) or 1:1 (1 × 10^6^ cfu/well) in the presence of antibiotics. Cytokines were measured in cell culture supernatants after 42 h of incubation (*n* = 5). Median and individual pigs shown. (**B**) Cytokine levels in supernatants of PBMC stimulated with *S. suis* strains 10 (*cps2*), 13-00283-02 (*cps7*) or 16085/3b (*cps9*) at the PBMC-to-bacteria ratios 10:1 and 1:1 in the presence of antibiotics. Cytokines were measured after 42 h of incubation (*n* = 3). (**C**) TNF-α and IL-6 production in presence of neutralizing anti-IL-10 antibody or with an isotype control after the stimulation of porcine PBMC with *S. suis* strain 10 at the same PBMC-to-bacteria ratios in the presence of antibiotics. IL-10 production was determined for reference. Cytokines from supernatants were measured after 42 h of incubation (*n* = 9–11). Graphs show pooled data from four independent experiments with seven to ten week old piglets. For statistical analyses, Mann Whitney test was performed (** *p* < 0.01).

**Figure 4 pathogens-09-00040-f004:**
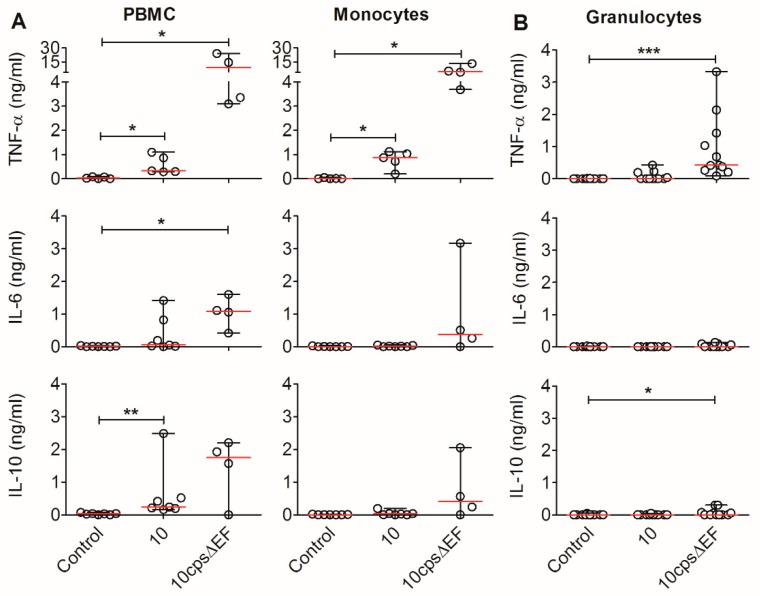
Monocytes are the main producers of TNF-α in response to encapsulated *S. suis.* (**A**) Levels of TNF-α, IL-6 and IL-10 secreted by PBMC and CD14-positive monocytes after stimulation with the wt *S. suis* strain 10 (*n* = 4–7) or the acapsular mutant strain 10cpsΔEF for 42 h in the presence of antibiotics. PBMC were cultivated at 1 × 10^6^ cells/well (200 µL) and monocytes were cultivated at 5 × 10^4^ cells/well. Bacteria were always used at 1 × 10^6^ cfu/well, which is equivalent to a PBMC-to-bacteria ratio of 1:1. Graphs show pooled data of four independent experiments with seven to 10 week old piglets. Depending on the resulting cell yield of the monocyte separation, fractions were stimulated with *S. suis* strain 10 and/or 10cpsΔEF. (**B**) Density gradient-purified granulocytes (1 × 10^6^ cells/well) from different donors, but pigs of the same herd (*n* = 11) were stimulated under the same conditions as the PBMC and the monocytes. The cytokines were quantified from cell culture supernatant by ELISA. Graphs show data from two independent experiments with seven to 10 week old piglets. * *p* < 0.05; *** p* < 0.01; **** p* < 0.001 according to Mann Whitney test of control (medium) versus *S. suis*-stimulated samples.

**Figure 5 pathogens-09-00040-f005:**
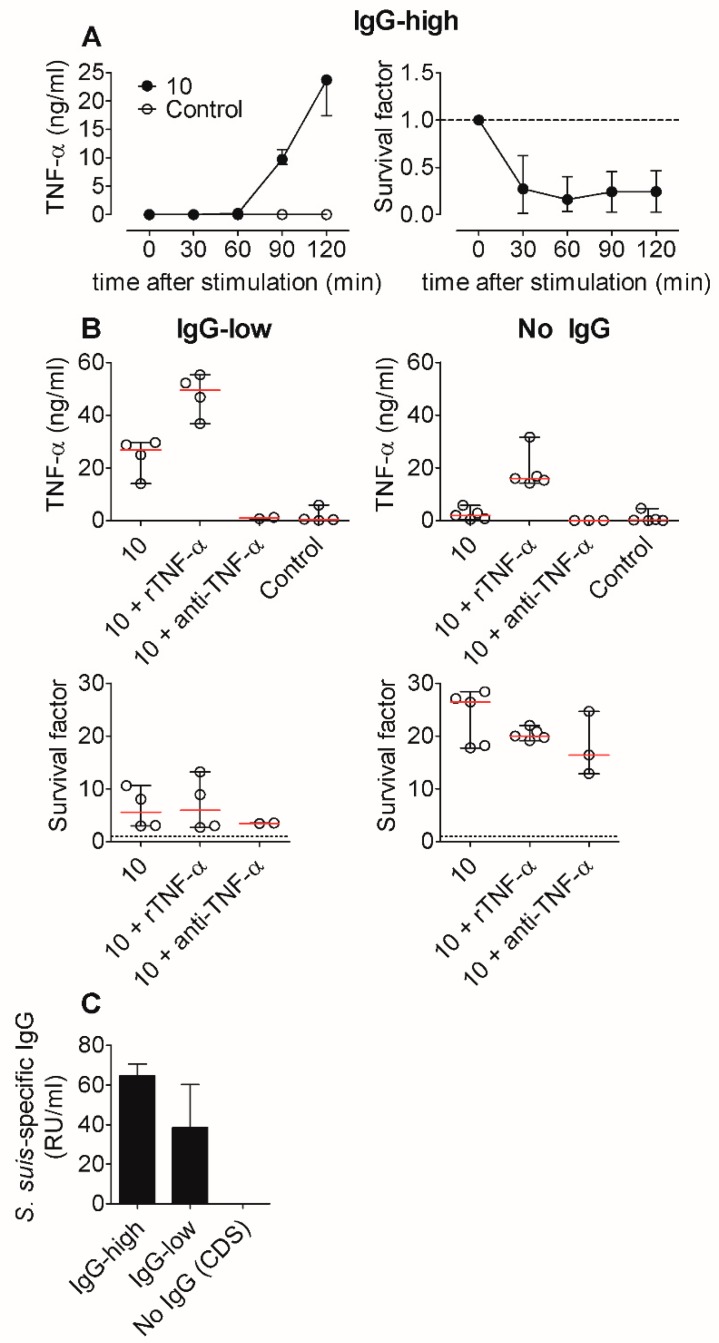
TNF-α does not contribute to bacterial killing in vitro, not even in the absence of *S. suis*-specific antibodies. (**A**) Kinetics of TNF-α induction and bacterial survival after infection of whole blood with 2 × 10^6^ cfu/mL *S. suis* strain 10 (*n* = 3). (**B**) Blood samples of piglets with low specific IgG (*n* = 2–4) or blood cells reconstituted with colostrum-deprived serum (CDS, see Section 4.2 Methods) (no specific IgG; *n* = 3–5) were infected with the viable *S. suis* strain 10 alone or additionally treated with porcine recombinant TNF-α (rTNF-α, 10 ng/mL) or neutralizing anti-TNF-α antibody (4 µg/mL). Bacterial survival was determined after 2 h. (**C**) Relative units per mL (RU/mL) of *S. suis* strain 10-specific IgG in sera of different groups of piglets. RU were calculated relative to a reference serum. Blood and sera were categorized based on the amount of *S. suis* strain 10-specific IgG: IgG-high > 60/mL RU (*n* = 3); IgG-low < 60 RU/mL (*n* = 4); No IgG = 0 RU/mL, using CDS with no detectable *S. suis* strain 10-specific IgG antibodies for the reconstitution of porcine blood cells. Graphs show data of four independent experiments with six to eight week old piglets.

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
