# Peer review of "Analysis of Porcine Pro- and Anti-Inflammatory Cytokine Induction by S. suis In Vivo and In Vitro"

_pathogens, 2020, doi:10.3390/pathogens9010040_

Round 1

Reviewer 1 Report

This manuscript presents a study on induction of pro-inflammatory cytokines TNF-alpha and IL-6, and the anti-inflammatory cytokine IL-10, upon infection with Streptococcus suis, in 7-10-week-old piglets, in an in vivo infection experiment, and in in vitro assays. In vivo, upon infection with S.suis strain 10 (3x10^8 cfu), three of six pigs developed pronounced bacteremia and two of these manifested an increase in IL-6, and two of these showed an increase in IL-10 (one animal showed an increase of both cytokines). In the first in vitro experiment a whole blood assay, claimed to mimic bacteremia in vivo, was applied: early induction of TNF-alpha and IL-6 was documented, and somewhat later induction of IL-10. To avoid confounding in this assay by bacterial proliferation, a cell culture of isolated blood mononuclear cells (obtained from healthy weaning piglets) with antibiotics was applied: the best induction of IL-6 and IL-10 was observed for an mutant acapsular bacterial strain, while TNF-alpha was strongly induced. Indications were given that inhibition of IL-10 results in enhanced induction of TNF-alpha and IL-6. Addressing the cell source of cytokines, separated subsets were tested: it was documented that monocytes  are the main producers of TNF-alpha upon stimulation with S.suis. In a kinetic experiments using the whole blood assay it was observed that TNF-alpha seems not required for bacterial killing as its production starts later, after kill of bacteria in the presence of IgG antibody.

The study seems interesting and the design and data presentation is clear. The quality of figures is fine. There are a number of points to be considered in revision of the manuscript.

The impact of the study for the general health of pigs (e.g., the pig industry) is not clear, leaving the reader in a limbo when thinking about the relevance of the work. The structure of the report, with the Methods section at the end, affects readability, e.g., the topic of work with piglets before weaning is not clear at first reading of the Results: it is advised to bring more source data to enhance the insight to the first sections of the Results. There should be a rationale provided for selection of young piglets, including piglets just before weaning, in this study. There should be a rationale provided for the selection of the three cytokines for this study. There are large series of many pro-inflammatory and anti-inflammatory cytokines. Normaly, spectra of cytokines are analysed and not selected small-sized series. In this respect the study design seems somewhat outdated. There should be a rationale provided for the dose selection in the in vivo infection experiment. The question is raised whether a higher dose results in more animals with pronounced bacteremia, and a lower dose in more animals with controlled bacteremia. Noteworthy, the three animals with controlled bacteremia are not informative for the purpose of this study; not only because there was no increase in cytokines in the circulation, but also since there is no link between concentrations in in vitro experiments and dose in the in vivo experiment. In the presentation of experimental protocols and data, the basis of replicates should be given, i.e., are replicated from the same original source like a single blood donor, or from different sources? In individual measurements in a distinct assay, how many replicates from a single sample were used? Etc, … In the in vivo study it seems to be clear in Figure 1 that measurements at different time points were done in the same single animal: the question is raised regarding animal welfare, i.e. whether it is allowed to bring piglets four times within 24 hours under full anesthesia. In Figure 5 it appears that the time-response is from the same culture, which raises the question to what extent the reduction in volume (1) affects the outcome of the experiment, and (2) is taken into consideration in calculation of the outcome. The material abbreviated as colostrum deprived serum (CDS) needs clarification, amongst others with respect to the source. How were animals grown up without access to colostrum? What was the level of antibodies to S.suis in these animals? It is up to speculation whether CDS manifests other abnormalities than only the claimed absence of maternally-derived antibodies to S.suis. Noteworthy, colostrum deprivation mainly provides immunity at the mucosa-associated lymphoid system. The discussion is rather long, and does not address the relevance of the study, and the relevance of the outcome of experiments. It is rather a literature review than a proper discussion of data.

Author Response

Changes in the manuscript relating to comments by reviewer 1 have been marked yellow.

Changes in the manuscript deemed necessary by the authors or changes relating to comments by both reviewers have been marked blue.

Reviewer 2 Report

The authors investigated the cytokines profile induced upon S. Suis infection: circulating levels of pro-inflammatory TNF-a and IL-6 and anti-inflammatory IL-10 were investigated 13-19 hours post-infection. In vitro studies were also performed, and monocytes were identified as the source of these cytokines in response to Streptococcus suis infection. In addition, the effect of TNF-a on bacterial killing was investigated and the authors analyzed differences between a Streptococcus suis wild type (wt) and an acapsular mutant.

In this work experiments were carefully planned and results are accessibly presented and thoughtfully discussed.  The information generated by this study should be of great interest to researchers working on Streptococcus suis, since a better understanding of the pathogenetic mechanisms of this bacteria can improve the development of control strategies.

Nevertheless, there are some moderate comments that would need to be addressed.

The authors should clarify more why the circulating levels of these cytokines during Stresptococcus suis infection were investigated. Why levels of TNF-a, IL-6 and IL-10 were monitored and not circulating levels of IL-8, IFN-g or other cytokines? Figure legend 1: replace ’13-19’with ‘0, 13, 16, 19’ (line 82). Figure legend 2: add ‘in vitro’, otherwise it is not clear (line 105). Figure 4. Differences in the TNF-a release from PBMC and monocytes in response to Streptococcus suis WT are not clear, because the TNF-a release in response to the acapsular strain is very high. I personally think that these differences should be highlighted in an additional figure or table. I personally think that it would have been better to show differences between the cytokine response of PBMC and PBMC depleted of the CD14 fraction. Discussion.

o   I think that the discussion would benefit from a brief introduction.

o   The authors concluded from the in vitro experiment that TNF-a did not contribute to the killing of this pathogen, nevertheless authors can’t rule out that in vivo this cytokine might contribute indirectly to bacterial killing, activating host-defense mechanisms not reproduced in vitro.

o   In addition, TNF-a levels early post-infection were not assessed, and this should be highlighted in the discussion.

o   I think that line 253-254 should be rephrased.

o   Line 263: replace ‘blood’ with ‘leukocytes’.

Material and methods.

o   Line 348: TNF-a, IL-6, IL-10 levels at 0 h post-infection were also determined.

o   4.4. Please explain more how granulocytes were purified and how their purity was assessed.

o   4.5 Provide information about monocytes and granulocytes.

Author Response

Changes in the manuscript relating to comments by reviewer 2 have been marked green.

Changes in the manuscript deemed necessary by the authors or changes relating to comments by both reviewers have been marked blue.

Round 2

Reviewer 1 Report

The authors have carefully addressed the comments made on the original manuscript, and have changed the manuscript accordingly. The detailed response to the comments made is highly appreciated.

Author Response

Thanks for your reviewing.
